# SimuPhy: Towards Physical Understanding, Reasoning, and Evaluation via Code Generation

## Abstract

Large language models (LLMs) have achieved remarkable progress in mathematics and code generation, yet their ability to reason about the physical world remains underexplored. Unlike mathematical reasoning, which can be expressed symbolically in text, physical reasoning is inherently tied to motion and dynamic processes. In this paper, we present SimuPhy, a novel task and dataset for evaluating LLMs' understanding, reasoning, and coding-based representation of physical laws. In SimuPhy, a model is given a motion description and tasked with generating code that simulates it. The resulting simulation is executed into a video, which is then evaluated by a vision–language model with predefined verification questions. SimuPhy contains 7,625 motion descriptions, including a curated 300-example test set with human verification. We evaluated 10 advanced LLMs, and find that even the strongest model, Deepseek-671B achieves only 20.6% pass rate, highlighting the difficulty of the task and the lack of physical law reasoning in current models. Building on this setup, we explore reinforcement learning with verifiable rewards (RLVR), pairing it with supervised fine-tuning (SFT) to improve models' ability to generate physically consistent simulations. Together, SimuPhy and our verifiable reward training pipeline provide a foundation for bridging language models toward genuine physical understanding.

## 1 Introduction

Recent frontier reasoning models such as GPT-o3 (OpenAI, 2025), DeepSeek-R1 (DeepSeek-AI, 2025), and Qwen3 (Qwen, 2025) have demonstrated impressive performance and generalizability across diverse domains, particularly in mathematical reasoning and code generation. These advances suggest that large language models (LLMs) are capable of complex reasoning and problem-solving when trained with appropriate supervision.

However, their ability to understand and simulate the *physical laws* remains underexplored. When asked to generate code for physical simulations (e.g., *simulate how a stone ball falling into a container filled with water and bouncing inside it*), existing models often fail in three critical aspects: (1) producing executable, error-free code; (2) generating physically consistent trajectories, such as preventing objects from penetrating solid boundaries or floating unnaturally; and (3) aligning the simulation with textual descriptions (e.g., the prompt specifies that the ball should "bounce", but the output shows it sinking or remaining static). These shortcomings reveal a fundamental limitation of current LLM training and evaluation: models are trained primarily on textual signals, even in domains like physics and dynamics, and are typically evaluated only for textual correctness rather than physical plausibility.

In this paper, we introduce **Simulation Physics (SimuPhy)**, a novel task and dataset for evaluating and improving LLMs' understanding, reasoning, and coding-based representation of physical laws. The core idea is to require models to comprehend textual descriptions of physical scenarios and generate executable simulations that reproduce the corresponding dynamic processes. To construct the dataset, we propose an automatic pipeline: starting from 52 physical concepts across five domains, we use GPT-4o (OpenAI, 2024) to generate basic scenarios that are progressively enriched with additional conditions to control difficulty. Each scenario is validated for physical plausibility,

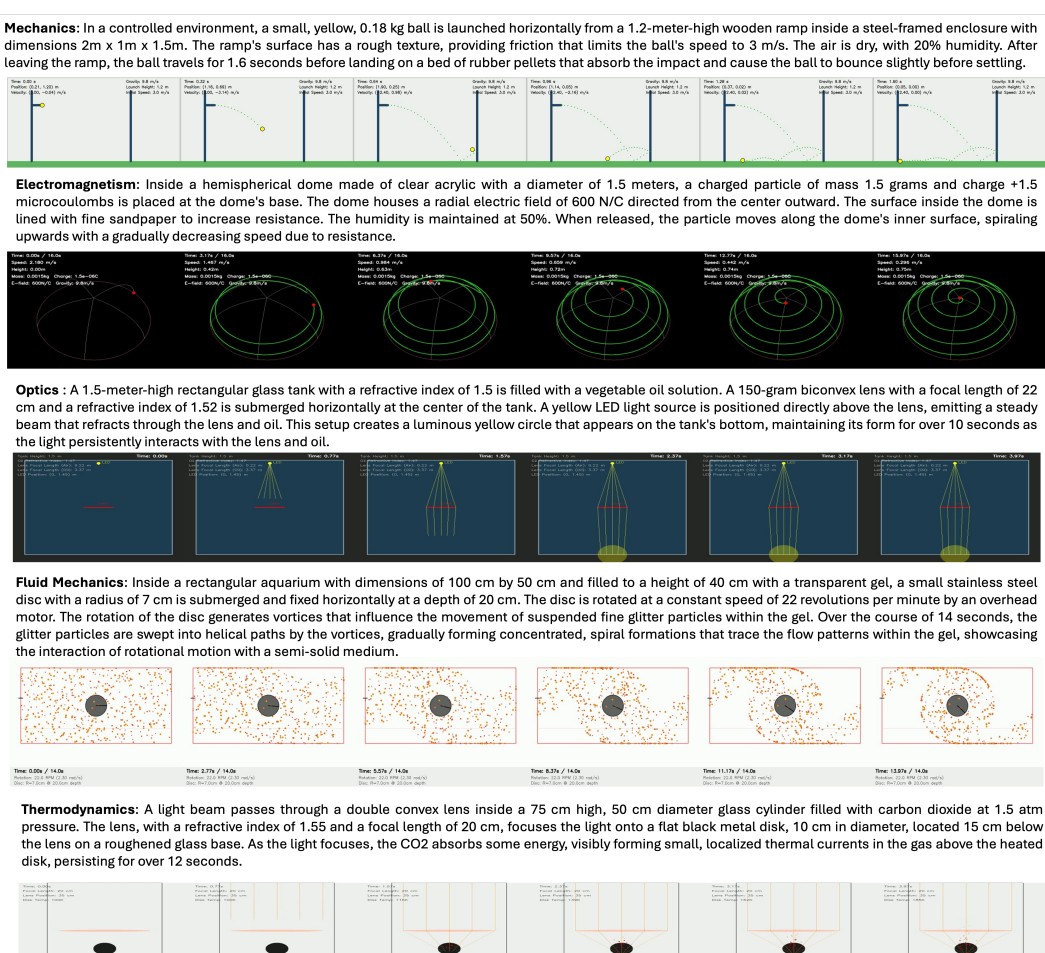

**Mechanics:** In a controlled environment, a small, yellow, 0.18 kg ball is launched horizontally from a 1.2-meter-high wooden ramp inside a steel-framed enclosure with dimensions 2m x 1m x 1.5m. The ramp's surface has a rough texture, providing friction that limits the ball's speed to 3 m/s. The air is dry, with 20% humidity. After leaving the ramp, the ball travels for 1.6 seconds before landing on a bed of rubber pellets that absorb the impact and cause the ball to bounce slightly before settling.

**Electromagnetism:** Inside a hemispherical dome made of clear acrylic with a diameter of 1.5 meters, a charged particle of mass 1.5 grams and charge +1.5 microcoulombs is placed at the dome's base. The dome houses a radial electric field of 600 N/C directed from the center outward. The surface inside the dome is lined with fine sandpaper to increase resistance. The humidity is maintained at 50%. When released, the particle moves along the dome's inner surface, spiraling upwards with a gradually decreasing speed due to resistance.

**Optics :** A 1.5-meter-high rectangular glass tank with a refractive index of 1.5 is filled with a vegetable oil solution. A 150-gram biconvex lens with a focal length of 22 cm and a refractive index of 1.52 is submerged horizontally at the center of the tank. A yellow LED light source is positioned directly above the lens, emitting a steady beam that refracts through the lens and oil. This setup creates a luminous yellow circle that appears on the tank's bottom, maintaining its form for over 10 seconds as the light persistently interacts with the lens and oil.

**Fluid Mechanics:** Inside a rectangular aquarium with dimensions of 100 cm by 50 cm and filled to a height of 40 cm with a transparent gel, a small stainless steel disc with a radius of 7 cm is submerged and fixed horizontally at a depth of 20 cm. The disc is rotated at a constant speed of 22 revolutions per minute by an overhead motor. The rotation of the disc generates vortices that influence the movement of suspended fine glitter particles within the gel. Over the course of 14 seconds, the glitter particles are swept into helical paths by the vortices, gradually forming concentrated, spiral formations that trace the flow patterns within the gel, showcasing the interaction of rotational motion with a semi-solid medium.

**Thermodynamics:** A light beam passes through a double convex lens inside a 75 cm high, 50 cm diameter glass cylinder filled with carbon dioxide at 1.5 atm pressure. The lens, with a refractive index of 1.55 and a focal length of 20 cm, focuses the light onto a flat black metal disk, 10 cm in diameter, located 15 cm below the lens on a roughened glass base. As the light focuses, the $CO_2$ absorbs some energy, visibly forming small, localized thermal currents in the gas above the heated disk, persisting for over 12 seconds.

Figure 1: Representative text–to–code–to–video examples from the SimuPhy benchmark. Each video is rendered from LLM-generated code and illustrates the described dynamic physical process.

paired with verification questions describing expected motion, and further refined using DeepSeek-R1-0528 (DeepSeek-AI, 2025) to cross-validate data quality (more details in § 3).

This pipeline yields the SimuPhy dataset, comprising 7,625 dynamic scenarios across five major domains of physics: mechanics, electromagnetism, optics, fluid mechanics, and thermodynamics. Each scenario centers on one core concept, with one to three additional conditions controlling difficulty. To support reliable evaluation, we also include a curated test set of 300 scenarios with mannual verification. Representative examples and their corresponding video simulations are shown in Fig. 1.

We evaluate 10 state-of-the-art LLMs on SimuPhy and find that the best-performing model achieves only a 20.6% pass rate (average@8). These results demonstrate not only that existing LLMs fall short in reasoning about and simulating physical phenomena, but also the effectiveness of SimuPhy as a challenging benchmark for probing physical reasoning.

Finally, we introduce a new vision-verifiable reward signal for reinforcement learning. Using this reward, we train DeepSeek-R1-Distill-Qwen-7B (DeepSeek-AI, 2025). Experimental results show that models trained with SimuPhy under this setup achieve substantially better alignment between textual descriptions and simulated motions. Notably, our trained 7B model attains performance competitive with the much larger Qwen3-32B, highlighting the effectiveness of both our dataset and training method. To the best of our knowledge, this is the first work to leverage visual verification signals for training LLMs in physical reasoning. To summarize, our contributions are three-fold:

1. We present **SimuPhy**, a novel task and dataset for evaluating LLMs' physical understanding, reasoning, and code-based simulation ability.

2. We evaluate 10 frontier LLMs, showing even the strongest achieves only 20% pass rate, highlighting both model limitations and the effectiveness of SimuPhy.

3. We propose a novel reinforcement learning approach with visual verification rewards, and demonstrate that, using our data and method, a 7B model can match the performance of a 32B model.

## 2 RELATED WORK

**Reasoning Evaluation for LLMs.** Recent research enhance LLMs' reasoning capabilities by equipping them with complex thinking before reaching final responses. Such LLMs include Ope-nAI's o-series (OpenAI, 2024; 2025), DeepSeek-R1, V3.1 (DeepSeek-AI, 2025; 2024), K2-Think (Cheng et al., 2025) and Qwen3 (Qwen, 2025). As the reasoning performance continue to advance, how to benchmark such capabilities become a vital problem. Previous study focus on math datasets, including AIME (MAA, 2024) and MATH (Hendrycks et al., 2021). Some research focus on challenging coding tasks, such as SWEBench (Jimenez et al., 2024) and LiveCodeBench (Jain et al., 2024).

**Physical Reasoning with LLMs** In contrast to math and code, physical reasoning requires understanding how objects behave under dynamic laws of motion. Early benchmarks addressed this through static attributes and plausibility judgments (e.g., elasticity, hardness, brittleness), often framed as multiple-choice or scenario-based QA (Zellers et al., 2018; Bisk et al., 2019; Wang et al., 2023). Moving beyond static attributes, more recent benchmarks emphasize explicit physics problem solving, ranging from undergraduate and Olympiad-level questions to robustness-focused and principle-based evaluations (Xu et al., 2025a; Yu et al., 2025; Zhang et al., 2025; Xu et al., 2025b; Qiu et al., 2025). Other approaches integrate simulators for analysis-by-synthesis reasoning (Cherian et al., 2024). While these benchmarks provide valuable insights into textual reasoning, they stop short of validating full end-to-end simulations.

**Vision–Language Benchmarks.** Complementary work has examined physics through perceptual grounding. SeePhys (Xiang et al., 2025) benchmarks diagram-centric, vision-essential QA, while PhysBench (Chow et al., 2025) evaluates multimodal understanding from videos, images, and text. Outside AI, interactive platforms such as myPhysicsLab[1] illustrate the feasibility of simulation-driven reasoning but are not designed around LLMs and lack textual task descriptions.

**Reinforcement Learning from Vision Signal** While reinforcement learning from vision signals has been extensively studied in domains such as robotics (Zitkovich et al., 2023), visual navigation (Wang et al., 2019), and embodied decision-making (Huang et al., 2025), these efforts focus on improving performance in vision-centric tasks. In contrast, we introduce the first **code-video-VLM verification closed-loop paradigm** for language model training. Our approach leverages a vision–language model (VLM) as a judge, evaluating not only textual reasoning accuracy but also the full dynamic process governed by physical laws.

SimuPhy differs from all prior approaches by coupling text-to-code generation with visual verification, validating full simulated rollouts, and enabling automatic judgments without ground-truth labels. In doing so, it shifts evaluation from isolated textual answers to end-to-end physical reasoning, providing a new foundation for bridging LLMs and the physical world.

## 3 SIMUPHY DATASET AND BENCHMARK

In this section, we first present the details of **SimuPhy** dataset construction, and then introduce the evaluation protocol, followed by the dataset statistics.

### 3.1 DATASET CONSTRUCTION

**Domain Selection** Our goal is to construct a dataset that encompasses a broad range of visualizable physical concepts. We first select five general classical physics domains—mechanics, electro-

---

[1] https://www.myphysicslab.com/

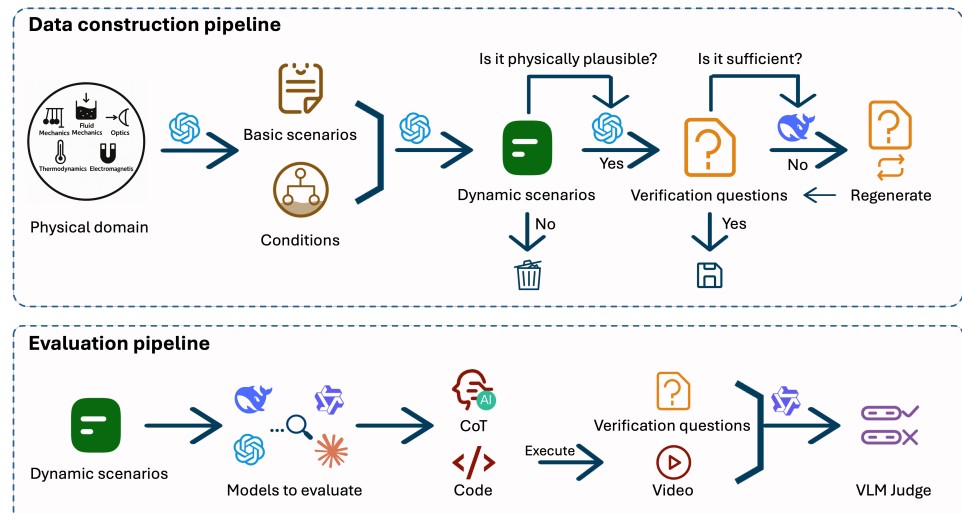

Figure 2: Overview of the SimuPhy dataset construction process, including dynamic scenario and visual question generation, as well as scenario–reasoning trace–code consistency assessment.

magnetism, optics, fluid mechanics, and thermodynamics. To provide both breadth and structure, we then adopt a two-level taxonomy, consistent with standard categorizations of classical physics (Karaoglu, 2020), from which we select 52 concepts such as fluid dynamics, angular momentum, and gravity. The detailed information is shown in Fig. 4a and § A.3.

**Scenario & Verification Question Generation**   The data construction process is depicted in Fig. 2. Starting from the 52 concepts, we first use GPT-4o (OpenAI, 2024) to generate *basic scenarios* (e.g., "a spinning disk on a frictionless table"). These scenarios are then enriched with additional *conditions*, such as collisions, external forces, or non-uniform properties, where the number of conditions determines the difficulty level. Each candidate scenario is checked for physical plausibility by judging whether it is possible to exist. For those deemed valid, GPT-4o generates a set of *visual verification questions* that describe the expected motion. We explicitly instruct the model to phrase these as questions about a video, ensuring that if all questions are answered as True, the video can be judged to faithfully represent the intended scenario.

**Quality Improving**   To strengthen quality control further, we introduce a second layer of review: We utilize DeepSeek-R1-0528 (DeepSeek-AI, 2025) to evaluate the generated verification questions, identifying errors, ambiguities, or omissions. When issues are detected, it revises the questions to improve clarity, correctness, and completeness. This multi-step pipeline, with plausibility filtering and independent review, ensures that the resulting scenarios and verification questions meet a high standard of reliability and data integrity. Fig. 3 shows an example of the data.

In this way, we construct a dataset comprising 7,625 validated scenarios with their associated verification questions, of which 7,325 are allocated for training and 300 are reserved for evaluation.

**Training Data Generation**   For the entire dataset, we further generate eight candidate responses for each scenario using DeepSeek-R1-0528 (DeepSeek-AI, 2025). Each response is automatically annotated with a correctness label by our evaluation pipeline, which we describe in detail in § 3.2. This procedure yields a diverse collection of model outputs paired with fine-grained correctness signals, making the dataset suitable for both supervised fine-tuning (SFT) and reinforcement learning (RL) approaches. In total, we obtain 12,556 validated responses spanning 4,182 scenarios, which can be directly used for SFT. The remaining 3,143 scenarios, for which no valid responses are available, are still valuable for RL training. By releasing this dataset to the community, we aim to enable broader research on aligning large language models with verification-centric reasoning tasks.

---

**Domain:** Electromagnetism     **Concept:** Electric Potential

---

**Basic Scenario:** A charged particle performs **spiral motion** in an electric field.

---

**Conditions:** 1. Introducing **conductors of different shapes** to change the electric field distribution; 2. Including **surface resistance or friction** to gradually dissipate the particle's kinetic energy.

---

**Dynamic Scenario:** Inside a **hemispherical dome** made of clear acrylic with diameter **1.5m**, a charged particle of mass **1.5g** and charge **+1.5μC** is placed at the dome's base. The dome houses a radial electric field of **600N/C** directed from center outward. Surface lined with fine sandpaper for resistance. Humidity: **50%**. When released, particle moves along dome's inner surface, **spiraling upwards** with **gradually decreasing speed** due to resistance.

---

**Verification Questions:** 1. A **hemispherical dome** is present as container; 2. A **charged particle** is initially placed at dome's base; 3. Charged particle moves along **inner surface** of dome; 4. Particle moves in **spiraling trajectory**; 5. Particle's **speed visibly decreases** as it spirals upwards; 6. Particle takes **visible time** to reach top; 7. Particle **reaches the top** of dome after motion.

---

Figure 3: Example from SimuPhy. A corresponding video is shown as the second entry in Fig. 1.

## 3.2 Code–Video–VLM Evaluation Pipeline

We propose a novel evaluation pipeline for assessing LLM-generated simulations, which consists of three steps: *(i)* First, the tested LLM is given a scenario description together with an instruction prompt that asks it to generate Python code simulating the scenario. The prompt specifies key details such as how to represent objects, the duration of the video, and the requirement to save the output in MP4 format (the exact prompt is shown in § G). *(ii)* Then, we collect the model's response, extract the Python code enclosed in python code blocks and execute it to render videos depicting the corresponding dynamic motions. *(iii)* Finally, we feed the generated videos, along with the scenario description and associated verification questions, into the Qwen-2.5-72B-VL model.[2] We prompt it to answer each verification question with True, False, or Not Sure, accompanied by a confidence score from 1–5. A scenario is judged consistent if all answers are True, or if fewer than three Not Sure answers have confidence scores below 4. Low-confidence Not Sure responses are treated as correct due to ambiguity, while high-confidence ones are treated as incorrect, reflecting unclear video evidence. More visualized examples of VLM judgments, including both consistent and inconsistent cases, are provided in § A.3.1.

To validate the effectiveness of our proposed pipeline, we feed our test set to Deepseek-R1-0528 (DeepSeek-AI, 2025) with 8 responses per sample, yielding 2,400 responses in total, of which 2,189 videos were successfully rendered. Two annotators independently evaluated these videos, achieving an inter-annotator agreement of 87%, indicating strong consistency. After resolving disagreements through discussion, the final human labels were established. We then compared these labels against the outputs of our automatic VLM judgers. Among them, 2,030 cases were consistent, resulting in an overall accuracy of 93.2%. This high level of agreement demonstrates the reliability of our evaluation approach.

## 3.3 Analysis of Dataset Properties

In this section, we analyze the properties of the **SimuPhy** dataset, including domain distribution, token length distribution, and VLM verification counts.

We begin with domain distributions. Among the five general domains Fig. 4a, mechanics includes 24 core concepts and electromagnetism includes 16, together comprising the majority of the dataset. The remaining three domains account for only 12 core concepts in total. We focus on mechanics and electromagnetism because their concepts are more common in daily life and generally easier to understand.

---

[2]We compared several open-source vision–language models on our task and found that Qwen-2.5-72B-VL achieved the most accurate judgments.

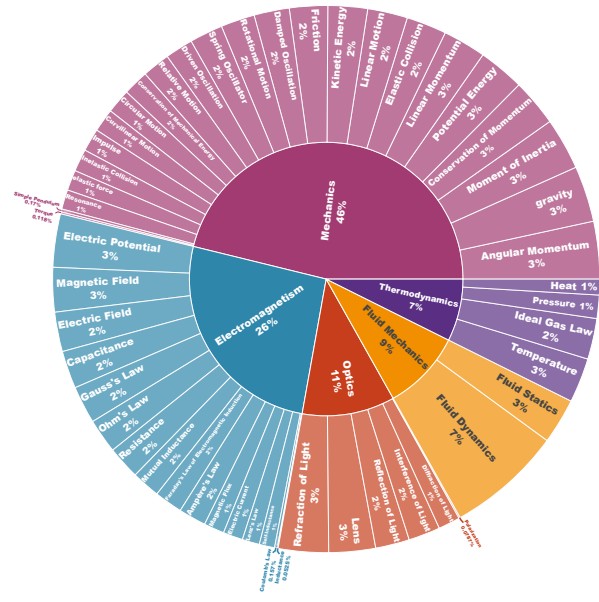

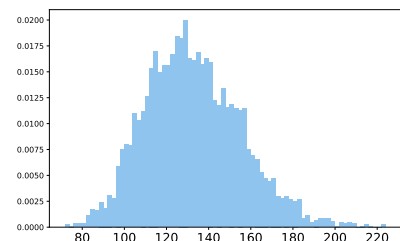

(b) Token count distribution per scenario in SimuPhy, measured using the Qwen3 tokenizer.

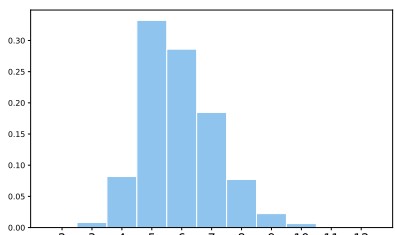

(a) Topic distribution of generated scenario across physics domains. Inner ring: domains; outer ring: concepts. Labels show the percentages.

(c) VLM verification question count distribution in SimuPhy.

Figure 4: Distributions of topics, scenario description token counts, and VLM verification questions in the SimuPhy dataset.

Beyond domains, we examine both the length of generated scenarios and the number of associated verification questions. Using the Qwen3 tokenizer (Fig. 4b), scenario lengths range from 72 to 224 tokens, with a near-normal distribution centered around 130, suggesting moderate complexity while avoiding extremes of brevity or verbosity. For verification questions (Fig. 4c), each scenario yields between 3 and 10, most often 5 to 7, which offers sufficient coverage of the motion process without redundancy while preserving scalability across varying levels of complexity.

## 4 EVALUATION OF FRONTIER LLMS

Table 1: Comparison of frontier LLMs on the SimuPhy benchmark. Metrics include Executable Rate (E.R.), Render Rate (R.R.), Playable Rate (P.R.), and VLM-evaluated Accuracy (Acc.). Avg@8 reports average performance across 8 runs, while Pass@8 evaluates at the scenario level, where a scenario is considered correct if at least one of the 8 generated codes yields a valid solution.

| Model | Size | Avg@8 | | | | Pass@8 | | | |
|---|---|---|---|---|---|---|---|---|---|
| | | E.R. | R.R. | P.R. | Acc. | E.R. | R.R. | P.R. | Acc. |
| Gemini-2.5-pro (Comanici et al., 2025) | - | 98.5 | 80.6 | 41.2 | 11.5 | 100.0 | 99.7 | 77.3 | 40.0 |
| Claude-3.7-sonnet (Anthropic, 2025) | - | 99.6 | 85.5 | 19.3 | 4.0 | 100.0 | 99.0 | 59.7 | 20.7 |
| GPT-o3 (OpenAI, 2025) | - | 95.2 | 72.3 | 64.1 | 13.4 | 100.0 | 100.0 | 96.3 | 43.0 |
| GPT-o4-mini (OpenAI, 2025) | - | 97.8 | 95.4 | 81.6 | 14.4 | 100.0 | 100.0 | 95.0 | 40.3 |
| GPT-oss-20b (OpenAI, 2025) | 21.5B | 98.5 | 86.8 | 49.3 | 7.7 | 100.0 | 100.0 | 93.3 | 27.0 |
| GPT-oss-120b (OpenAI, 2025) | 120B | 99.7 | 93.3 | 51.9 | 6.8 | 100.0 | 100.0 | 66.3 | 23.0 |
| Qwen3-32B (Qwen, 2025) | 32B | 99.9 | 84.7 | 72.1 | 10.5 | 100.0 | 99.3 | 97.7 | 34.7 |
| Qwen3-235B-A22B (Qwen, 2025) | 235B | 99.3 | 95.1 | 63.0 | 12.3 | 100.0 | 100.0 | 91.3 | 34.7 |
| DeepSeek-R1-0528 (DeepSeek-AI, 2025) | 671B | 99.4 | 96.8 | 90.8 | 20.6 | 100.0 | 100.0 | 100.0 | 54.7 |
| DeepSeek-V3.1(DeepSeek-AI, 2024) | 671B | 99.7 | 91.0 | 56.4 | 9.8 | 100.0 | 99.7 | 87.7 | 34.7 |

We evaluate a series of best reasoning models, including open-source models: GPT-oss (20B, 120B) (OpenAI, 2025), Qwen3 (32B, 235B-A22B) (Qwen, 2025), and Deepseek-R1-0528 (DeepSeek-AI, 2025) and Deepseek-V3.1 (671B)(DeepSeek-AI, 2024), as well as the proprietary models include

Gemini-2.5-pro (Comanici et al., 2025), Claude-3.7-Sonnet (Anthropic, 2025), GPT-o3 (OpenAI, 2025), and GPT-o4-mini(OpenAI, 2025). We use the evaluation protocol described in § 3.2.

**Metrics**   Based on the evaluation process, we designed the following metrics to measure LLMs capability on our benchmark.

- **Executable Rate (E.R.)** The percentage of executable code among all the generated responses. This measure coding competence and environment readiness of LLMs. i.g. syntax, imports, API signatures, dependency handling, path handling, basic resource setup.

- **Render Rate (R.R.)** The percentage of codes that successfully rendered videos among all the generated responses. It assesses the capability of runtime reliability and pipeline assembly. i.g. correct sequencing of simulation and rendering steps, resource and state management, file IO robustness, error handling during long runs.

- **Playable Rate (P.R.)** The percentage of playable videos among all the generated responses.

- **Accuracy (Acc.)**  The percentage of VLM evaluated pass among all the generated responses. The measures the end-to-end performance of LLMs, which combines the physical understanding, reasoning, and code simulation capability.

**Results**   Table. 1 summarizes the evaluation results of frontier LLMs on the SimuPhy benchmark. We report **Avg@8** and **Pass@8** to assess both the overall reliability of model outputs and their best-case capability per scenario. As shown in the table, several performance patterns emerge across executability, renderability, playability, and end-to-end accuracy.

First, all the models achieve very low performance in end-to-end evaluation, highlighting their limited ability to comprehensively understand physics and translate this understanding into code simulations. Notably, even the strongest model, DeepSeek-R1-0528, achieves only 20.6% average@8 performance. This underscores the exceptional difficulty of our benchmark.

Second, most models achieve high executable rate (E.R.) and render rate (R.R), which means that they are good at following instructions to generate code and save the video. The most drop happen in playable rate (P.R.) and accuracy (ACC.), revealing persistent challenges in translating physical phenomena into coherent videos and representing them correctly through code.

Third, we observe that different models exhibit distinct behaviors across the evaluation metrics. For instance, Qwen3-32B achieves a relatively higher playable rate compared to GPT-o3, but attains a lower final accuracy, suggesting that although Qwen produces more videos that can be executed and played, its simulations are less often aligned with the physical descriptions.

To better understand the difficulty of SimuPhy across domains, we present in Table. 2 a domain-level breakdown of DeepSeek-R1-0528's performance. The results show marked variation across physics domains. The model performs strongest on mechanics and fluid mechanics, achieving 25.8% Avg@8 in mechanics and 72.0% Pass@8 in fluid mechanics. By contrast, optics emerges as the most challenging domain, with only 9.3% Avg@8 and 31.4% Pass@8, likely reflecting the scarcity of optics-related supervision during training. Nevertheless, the generally low Avg@8 scores across all domains highlight the fundamental difficulty of physical law understanding and simulation for current LLMs.

Table 2: DeepSeek-R1-0528 performance breakdown of physical domains.

| Category | Avg@8 | Pass@8 |
|---|---|---|
| Mechanics | 25.8 | 60.6 |
| Electromagnetism | 16.0 | 51.4 |
| Optics | 9.3 | 31.4 |
| Fluid Mechanics | 23.5 | 72.0 |
| Thermodynamics | 17.2 | 45.8 |

## 5   MODEL TRAINING FOR BETTER PHYSICAL SIMULATION

In this section, we use our provided training set to fine-tune an LLM, § 5.1 describes supervised fine-tuning, while § 5.2 introduces reinforcement learning, covering optimization and reward design. Unless otherwise stated, we use a shared system prompt and user template for both SFT and RLVR. The full texts are provided in Appendix § G.

## 5.1 Bootstrapping the Policy via SFT

We use the data with correct labels mentioned in § 3 to finetune DeepSeek-R1-Distill-Qwen-7B. We use the AdamW optimizer, with maximum learning rate set to $4 \times 10^{-5}$ and linearly scaled to 0. We train the model for 2 epochs with a batch size of 256. During training, all samples are truncated to a maximum of $16,000$ tokens. The training takes about 16 GPU hours.

## 5.2 Vision–Language Verification for RL

Given a motion description $x$, the policy $\pi_\theta$ produces a structured reasoning trace and an executable program $c$. Running $c$ deterministically (modulo rendering non-determinism) returns a video $\mathcal{V}$ or fails. A vision–language judge (VLM) then answers a set of verification questions $\mathcal{Q} = \{q_i\}_{i=1}^{M}$, designed to test whether $\mathcal{V}$ matches the physical behaviors entailed by $x$. For each $q_i$, the VLM yields a ternary label $y_i \in \{\text{TRUE}, \text{FALSE}, \text{NOTSURE}\}$ and a discrete confidence $s_i \in \{1, \dots, 5\}$. The VLM judge prompt is provided in § A.

**Ternary-to-binary resolution.** Following our dataset protocol (§ 3), we resolve NOTSURE by a confidence threshold $\tau = 4$:

$$\tilde{y}_i \;=\; \begin{cases} 1, & y_i = \text{TRUE} \;\; \text{or} \;\; (y_i = \text{NOTSURE} \wedge s_i < \tau), \\ 0, & y_i = \text{FALSE} \;\; \text{or} \;\; (y_i = \text{NOTSURE} \wedge s_i \geq \tau). \end{cases} \tag{1}$$

Low-confidence NOTSURE is treated as positive due to visual ambiguity, while high-confidence NOTSURE is treated as negative for lack of visible evidence. If compilation or execution fails, we set $\tilde{y}_i = 0$ for all $i$.

**Verifiable reward.** The scalar reward is the fraction of resolved positives:

$$r(c \,|\, x) \;=\; \frac{1}{M} \sum_{i=1}^{M} \mathbf{1}[\tilde{y}_i = 1]. \tag{2}$$

This reward is sparse yet *verifiable*: it depends only on the rendered video and the fixed question set $\mathcal{Q}$, requiring no human labels at training time and directly reflecting physical correctness.

We adopt Group Relative Policy Optimization (GRPO) (Shao et al., 2024) for model training, see more details in § A.1.

## 5.3 Results

We report results for the *DeepSeek-R1-Distill-Qwen-7B* model trained with our SFT and RLVR pipelines. Overall results are summarized in Table. 3, the base model cannot perform end-to-end simulation. SFT improves all metrics significantly.

Unless otherwise stated, we use the VLM-as-judge accuracy reward $r_{\text{acc}}$ (Eq. 6) as the default. Compared to a model trained only with SFT, adding RL boosts the render-to-play conversion (P.R./R.R.) from 0.38 to 0.94 (Avg@8). In practice, this means that most rendered videos become playable, showing that RLVR significantly improves runtime reliability and pipeline assembly. Consider Pass@8: under RLVR, performance approaches saturation with E.R. 99.7%, R.R. 99.7%, and P.R. 98.3%, while Acc. rises from 11.0% to 45.0% for a gain of about $4.1\times$.

Table 3: Performance of DeepSeek-R1-Distill-Qwen-7B across three stages: base model, after SFT, and after RL with different reward signal.

| Model | Avg@8 | | | | Pass@8 | | | |
|---|---|---|---|---|---|---|---|---|
| | E.R. | R.R. | P.R. | Acc. | E.R. | R.R. | P.R. | Acc. |
| Base Model | 27.4 | 0.0 | 0.0 | 0.0 | 90.7 | 0.3 | 0.0 | 0.0 |
| w/ SFT | 53.3 | 45.4 | 17.1 | 1.8 | 94.3 | 92.3 | 66.3 | 11.0 |
| w/ RL (VLM-as-judge, acc.) | 91.1 | 60.4 | 56.7 | **9.8** | 99.7 | 99.7 | **98.3** | **45.0** |
| *reward variants* | | | | | | | | |
| w/ RL (VLM-as-judge, binary) | **99.5** | **98.1** | **67.1** | 6.5 | **100.0** | **100.0** | 89.0 | 22.7 |
| w/ RL (LLM-as-judge) | 92.2 | 83.9 | 16.5 | 2.7 | **100.0** | **100.0** | 55.3 | 14.3 |

With our training, the 7B model achieves **45.0%** Pass@8 accuracy, second only to DeepSeek-R1-0528 at 54.7%, and exceeding GPT-o3, GPT-o4-mini, Gemini-2.5-pro, and other models. At the

sample-level, the model attains 9.8% Avg@8 accuracy, which is still lower than many other models. These results demonstrate both the effectiveness of our data, training recipe, and vision reward signal, and that there remains room for further improvement.

**Reward Variants.** We compare the proposed reward method against two alternative variants, using the same set of verification questions. The first variant also employs *VLM-as-judge*, but defines the reward as the fraction of correctly verified questions, yielding a score between 0 and 1. The second variant utilizes binary criterion in which a sample passes only when every resolved verification is positive. The Third variant uses *LLM-as-judge* (text-only), where the verification questions are combined with direct judgments of the generated code. Further details are provided in § B.2.

The *VLM-as-judge (accuracy)* reward is best aligned with our end-to-end simulation: it converts renders into playable videos most reliably (Avg@8 P.R./R.R. ≈ **0.94** vs. 0.68 for VLM-binary and 0.20 for LLM) and delivers the highest task accuracy (Pass@8 Acc **45.0**%). For *VLM-as-judge (binary)*, E.R./R.R. nearly saturate and P.R. is high—but its sparse signal under-rewards fine-grained correctness and plateaus at lower downstream accuracy (Pass@8 Acc 22.7%). *LLM-as-judge* is inexpensive and boosts render rates, yet without visual grounding it is prone to optimistic false positives and reward hacking, yielding weaker playability and accuracy.

As shown by the reward trajectories in Fig. 5, all the three judges yield steadilt improving signals during RLVR, but both *VLM-as-judge* variants produce larger and more faithful rewards than the *LLM-as-judge*. We find that this gap arises from two sources.

**(i) Modality fidelity.** The VLM observes the rendered video $\mathcal{V}$ and answers $\mathcal{Q}_x$ about object motion, contacts, and temporal relations; the text-only LLM inspects $c$ and must *infer* runtime behavior from static structure. This induces optimistic false positives (e.g., physically implausible interactions, premature occlusions, or off-camera motions that look "plausible" in code but fail visually).

**(ii) Execution observability.** In our pipeline, failures to compile/execute set all resolved la-

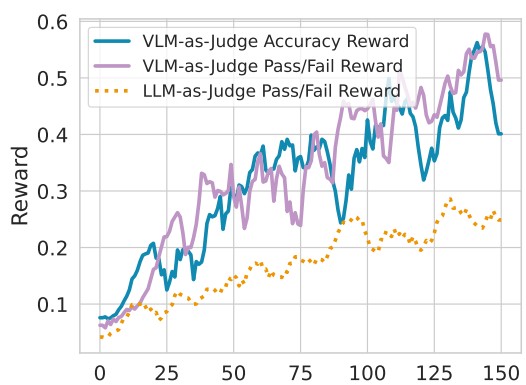

Figure 5: The reward curve during RL training. All three reward variants yield steadily improving signals.

bels to 0, tightly coupling VLM rewards to end-to-end reliability (E.R., R.R., P.R.). Consequently, the VLM rewards move in lockstep with render-to-play conversion (Table. 3), whereas the LLM judge answers $\mathcal{Q}'_x$ without running $c$ and therefore cannot detect many run-time or asset/camera issues; the policy can partially exploit this by emitting code that *looks* executable (imports, guards, placeholders) yet does not render a playable video.

Therefore, visual feedback closes the loop from code → video → VLM verification, aligning the reward with our end-to-end objective and making it harder to exploit spurious textual heuristics.

## 6 CONCLUSION

We introduced SimuPhy, a new task and benchmark that evaluates whether LLMs can understand, reason about, and simulate physical dynamics. We built a fully automatic construction pipeline that *(i)* generates topic-conditioned dynamic scenarios and visual verification questions, *(ii)* produces reasoning traces and executable code, and *(iii)* validates rendered videos via a VLM. The resulting dataset contains 7,625 scenarios across five physics domains, with a 300-example human-verified test set; VLM judgements show 93.2% agreement with human labels, supporting their use as verifiable signals. Benchmarking popular LLMs reveals that the task is challenging: even frontier models attain only modest accuracy under Avg@8, which highlights gaps in current physical-law reasoning. Building on SimuPhy, we employ reinforcement learning with verifiable rewards (RLVR) coupled with SFT, which substantially improves end-to-end text to code to video consistency; notably, our 7B model achieves performance competitive with frontier-scale systems.

## 7 REPRODUCIBILITY STATEMENT

To ensure the reproducibility of our work, we provide a detailed description of the **dataset construction pipeline** in § 3, covering scenario generation, code synthesis, and VLM-based verification. The complete **training and evaluation setup**, including model configurations, hyperparameters, and evaluation metrics, is reported in § 5, with additional ablation studies presented in the Appendix. All prompts used in our project are detailed in § G. Moreover, we include our code and scripts in the supplementary materials to facilitate replication. Together, these resources are intended to enable the community to reproduce our results and build upon SimuPhy for future research.

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

## A  IMPLEMENTATION DETAILS

### A.1  GRPO OPTIMIZATION

We adopt Group Relative Policy Optimization (GRPO) (Shao et al., 2024). For each prompt $x$, the old policy $\pi_{\theta_{\text{old}}}$ samples a *group* of $K$ candidates $c_{1:K} \sim \pi_{\theta_{\text{old}}}(\cdot \mid x)$ (e.g., $K = 8$). Executing each $c_k$ produces a reward $r_k = r(c_k \mid x)$ from Eq. equation 2. We compute a group-relative, length-invariant advantage

$$\bar{A}_k \;=\; \frac{r_k - \mu_x}{\sigma_x + \epsilon}, \qquad \mu_x = \frac{1}{K} \sum_{j=1}^{K} r_j, \;\; \sigma_x = \sqrt{\frac{1}{K} \sum_{j=1}^{K} (r_j - \mu_x)^2}, \tag{3}$$

and assign $\bar{A}_k$ uniformly to all tokens of $c_k$ to avoid length bias.

We then optimize a clipped likelihood-ratio objective *without* an explicit KL penalty, using asymmetric clipping to stabilize updates while preserving headroom for improvement:

$$\mathcal{L}_{\text{clip}}(\theta) = \frac{1}{K} \sum_{k=1}^{K} \frac{1}{|c_k|} \sum_{t=1}^{|c_k|} \min\!\big(\rho_{k,t}(\theta)\, \bar{A}_k,\; \text{clip}\big(\rho_{k,t}(\theta),\, 1 - \varepsilon_{\text{low}},\, 1 + \varepsilon_{\text{high}}\big) \bar{A}_k\big), \tag{4}$$

where

$$\rho_{k,t}(\theta) = \frac{\pi_\theta(c_{k,t} \mid x, c_{k,<t})}{\pi_{\theta_{\text{old}}}(c_{k,t} \mid x, c_{k,<t})}, \qquad \text{clip}(u, a, b) = \min\!\big(\max(u, a),\, b\big). \tag{5}$$

We maximize $\mathcal{L}_{\text{clip}}$ with Adam; $\varepsilon_{\text{low}} < \varepsilon_{\text{high}}$ (asymmetric clipping) curbs overly aggressive down-weighting while allowing moderate up-weighting of promising tokens. In our main run, we use Adam with a learning rate of $5 \times 10^{-7}$, train for 150 steps with a global train batch size of 32, totaling 576 GPU-hours.

### A.2  TRAINING DATA DETAILS

The SimuPhy dataset contains a total of 7,625 validated dynamic scenarios, of which 300 are reserved as a test set. For training, we use the remaining 7,325 examples. These are split into two parts:

- **Supervised Fine-Tuning (SFT):** 3,000 examples are used to fine-tune the base model with paired scenario descriptions, reasoning traces, and executable code.

- **Reinforcement Learning (RL):** the remaining 4,325 examples are used for reinforcement learning with verifiable rewards (RLVR), where VLM judgments provide reward signals.

This division ensures that the model benefits from both explicit supervised guidance and scalable reward-driven optimization.

### A.3  DATASET

We use a corpus of **7,625** samples covering five core physics domains with **52** fine-grained concept (see Table. 4). Mechanics contributes the largest share (3,520; 46.2%), followed by Electromagnetism (1,984; 26.0%), Optics (836; 11.0%), Fluid Mechanics (724; 9.5%), and Thermodynamics (561; 7.4%). Across concepts, the mean count is 146.6 samples (median 149.5; s.d. 82.6), indicating a moderately imbalanced, long-tailed distribution.

At the concept level, *Fluid Dynamics* is the largest category (519; 6.8%), with other frequent topics including *Angular Momentum* (259; 3.4%), *gravity* (252; 3.3%), and *Electric Potential* (240; 3.1%). On the sparse end, *Inductance* (4; 0.05%), *Polarization* (6; 0.08%), *Torque* (9; 0.12%), *Coulomb's Law* (12; 0.16%), and *Simple Pendulum* (13; 0.17%) have very few examples (five concepts have $< 20$ samples). The ten most common concepts together account for 33.7% of the data, whereas the ten rarest account for 5.8%.

Table 4: Distribution of samples across domains and concepts

| Domain | Concept | Number | Concept | Number |
|---|---|---|---|---|
| Electromagnetism | Electric Potential | 240 | Magnetic Field | 200 |
| | Electric Field | 178 | Capacitance | 170 |
| | Gauss's Law | 169 | Ohm's Law | 155 |
| | Resistance | 149 | Mutual Inductance | 128 |
| | Faraday's Law | 127 | Ampère's Law | 124 |
| | Magnetic Flux | 92 | Electric Current | 89 |
| | Lenz's Law | 78 | Self-Inductance | 69 |
| | Coulomb's Law | 12 | Inductance | 4 |
| | | | **Total** | **1984** |
| Fluid Mechanics | Fluid Dynamics | 519 | Fluid Statics | 205 |
| | | | **Total** | **724** |
| Mechanics | Angular Momentum | 259 | gravity | 252 |
| | Moment of Inertia | 234 | Conservation of Momentum | 213 |
| | Potential Energy | 208 | Linear Momentum | 194 |
| | Elastic Collision | 181 | Linear Motion | 180 |
| | Kinetic Energy | 178 | Friction | 172 |
| | Damped Oscillation | 159 | Rotational Motion | 153 |
| | Spring Oscillator | 150 | Driven Oscillation | 126 |
| | Relative Motion | 125 | Conservation of Mechanical Energy | 115 |
| | Circular Motion | 110 | Curvilinear Motion | 105 |
| | Impulse | 103 | Inelastic Collision | 101 |
| | Elastic Force | 92 | Resonance | 88 |
| | Simple Pendulum | 13 | Torque | 9 |
| | | | **Total** | **3520** |
| Optics | Refraction of Light | 226 | Lens | 210 |
| | Reflection of Light | 153 | Interference of Light | 143 |
| | Diffraction of Light | 98 | Polarization | 6 |
| | | | **Total** | **836** |
| Thermodynamics | Temperature | 201 | Ideal Gas Law | 184 |
| | Pressure | 105 | Heat | 71 |
| | | | **Total** | **561** |
| | | | **Grand total** | **7625** |

### A.3.1 SUPPLEMENTARY DATASET INFORMATION AND ILLUSTRATIVE EXAMPLES

As shown in Table. 4, the SimuPhy dataset spans 52 concepts, which are grouped into five major domains: Mechanics (3,520 samples), Electromagnetism (1,984), Optics (836), Fluid Mechanics (724), and Thermodynamics (561), resulting in a total of 7,625 scenarios. Mechanics and Electromagnetism dominate the distribution, reflecting their central role in classical physics and their higher executability and validation success rates during data generation.

We intentionally include both well-represented concepts (e.g., Angular Momentum, Electric Potential, Refraction of Light) and low-frequency ones (e.g., Inductance, Polarization, Torque) to ensure topic diversity. This long-tail coverage allows the benchmark to evaluate LLMs not only on common physical processes but also on challenging or visually subtle phenomena, providing a more comprehensive assessment of physical reasoning.

Moreover, we provide additional qualitative examples to further demonstrate the reliability of VLM judgments. As shown in Fig. 7, the pass cases correspond to scenarios where the LLM-generated code produces videos that are fully consistent with the textual descriptions. In contrast, Fig. 8 presents inconsistency cases flagged by the VLM (failure questions are highlighted in blue), where mismatches occur between the input description and the rendered simulation. These failure cases illustrate typical error modes in LLM-generated code, including missing or incomplete motions, object penetration through solid boundaries, and discrepancies between the described and simulated dynamics.

# B  VLM AS JUDGE VS. LLM AS JUDGE

In this section, we compare the performance of VLM as Judge and LLM as Judge to highlight the importance of our design. For the LLM as Judge setting, we send the code produced by DeepSeek-R1-0528 directly to the Qwen/Qwen2.5-72B-Instruct(Team, 2024) model and augment the evaluation with two additional questions: (1) whether the code can execute successfully, and (2) whether the code can generate and save a video. If either question receives a "False" response, the overall judgment for that item is marked as "False."

## B.1  AGREEMENT WITH HUMAN ANNOTATION

Both VLM as Judge and LLM as Judge use the same model responses. The only difference is that VLM as Judge executes the code, generates the corresponding video, and then passes the resulting video to Qwen/Qwen2.5-VL-72B-Instruct(Team, 2025), whereas LLM as Judge sends the code directly to Qwen/Qwen2.5-72B-Instruct(Team, 2024) without executing it. All other settings remain identical.

Table. 5 reports the results on the test set compared with human annotations. True/False means the model predicted "True" while the human annotated "False," and False/True means the model predicted "False" while the human annotated "True." Agreement reflects the consistency between model predictions and human annotations. As shown, VLM as Judge achieves a markedly higher agreement with human annotations (93.2%) than LLM as Judge (55.1%), demonstrating the benefit of incorporating video execution and rendering into the evaluation process.

Table 5: Evaluation Results under Different Judges. The table shows the comparison between model predictions and human annotations. For example, *True/False* means the model predicted True while the human annotated False, and *False/True* means the model predicted False while the human annotated True. *Agreement* indicates the consistency between model predictions and human annotations.

|  | True/True | False/False | True/False | False/True | Agreement |
|---|---|---|---|---|---|
| VLM as Judge | 398 | 1631 | 96 | 53 | 93.2% |
| LLM as Judge | 337 | 986 | 963 | 114 | 55.1% |

## B.2  REWARD DESIGN FOR RL TRAINING

We instantiate three reward functions to close the loop from code to video to judgment: two *VLM-as-judge* variants and one *LLM-as-judge* (text-only) variant. All VLM-based variants use the same question set $\mathcal{Q}_x = \{q_i\}_{i=1}^M$ prepared for prompt $x$ and the ternary-to-binary resolution rule in Eq. equation 1 (low-confidence NOTSURE counts as positive; high-confidence NOTSURE counts as negative). If compilation or execution fails, we set all resolved labels to 0 (negative).

**(1) VLM-as-judge: Accuracy reward.**  The VLM answers each $q_i$ about the rendered video $\mathcal{V}$ with $y_i \in \{\text{TRUE}, \text{FALSE}, \text{NOTSURE}\}$ and confidence $s_i \in \{1, \dots, 5\}$; we map to $\tilde{y}_i \in \{0, 1\}$ using Eq. equation 1. The reward is the fraction of resolved positives (cf. Eq. equation 2 in the main text):

$$r_{\text{acc}}(c \,|\, x) \;=\; \frac{1}{M} \sum_{i=1}^{M} \mathbf{1}[\tilde{y}_i = 1]. \tag{6}$$

This dense, verifiable signal correlates with process-level physical correctness while remaining inexpensive to obtain once $\mathcal{Q}_x$ is fixed.

**(2) VLM-as-judge: Pass/Fail reward.**  Using the same $\mathcal{Q}_x$ and resolution rule, we define a stricter binary criterion in which a sample is marked as *pass* only if all resolved answers are positive:

$$r_{\text{pass}}^{\text{VLM}}(c \,|\, x) \;=\; \mathbf{1}[\forall i \in \{1, \dots, M\}, \; \tilde{y}_i = 1]. \tag{7}$$

This variant provides a sparse, high-precision supervision signal that is useful for ablations and thresholded evaluation.

**(3) LLM-as-judge (text-only): Pass/Fail reward.** Here we do *not* execute $c$ nor render a video. Instead, we prompt a text-only LLM with *(i)* the motion description $x$, *(ii)* the generated code $c$, and *(iii)* an extended question set

$$\mathcal{Q}'_x \;=\; \mathcal{Q}_x \;\cup\; \{q_{\text{exec}}, q_{\text{video}}\},$$

where the two additional binary checks are:

- $q_{\text{exec}}$: *"Does the code execute without errors?"*
- $q_{\text{video}}$: *"Does the code generate and save a video?"*

The LLM is instructed to answer each question strictly with TRUE/FALSE (we parse case-insensitively and map *Yes/No* to TRUE/FALSE). Let $z_j \in \{\text{TRUE}, \text{FALSE}\}$ denote the LLM's answer to the $j$-th question in $\mathcal{Q}'_x$. The reward is

$$r_{\text{pass}}^{\text{LLM}}(c \,|\, x) \;=\; \mathbf{1}[\forall\, j,\; z_j = \text{TRUE}]\,. \tag{8}$$

Unlike the VLM-based variants, this setting relies solely on textual inspection of $c$ and may disagree with actual runtime behavior; see Table. 5 for an empirical comparison.

**Implementation notes.** For VLM judging we use the confidence threshold $\tau{=}4$ (Eq. equation 1) to resolve NOTSURE; however, the ambiguity gate is computed on the *raw* ternary labels ($y_i$) *before* resolution and applies regardless of confidences. We evaluate the gate first (i.e., if $n_{\text{ns}} > 2$, then $r_{\text{pass}}^{\text{VLM}}{=}0$), otherwise we proceed to check $\tilde{y}_i$.

Parsing and normalization follow the main text; failed compilation or rendering yields $r_{\text{acc}}{=}0$ and $r_{\text{pass}}^{\text{VLM}}{=}0$ by construction. The LLM-as-judge variant remains unaffected, as the text-only judge is instructed to answer strictly in TRUE/FALSE without a NOTSURE option.

## C  RESPONSE LENGTH

**Response Length for Tested Models** We further analyze the response lengths of different models on the SimuPhy test set, as shown in Fig. 6. The distribution reveals clear differences in reasoning styles across models. Compact models such as GPT-o4-mini and GPT-oss-120B typically generate concise responses (median $\sim$1k tokens), while larger models like Gemini-2.5-pro and Claude-3.7-sonnet tend to produce significantly longer outputs, often exceeding 3k tokens. Mid-sized models such as Qwen3-32B, Qwen3-235B, and DeepSeek-V3.1 fall between these extremes, with moderate but stable token usage.

Interestingly, longer responses do not necessarily correlate with higher accuracy on SimuPhy. For example, Gemini-2.5-pro produces the longest outputs yet achieves only moderate end-to-end accuracy (Acc. 11.5 in Avg@8), while DeepSeek-R1-0528 generates relatively shorter responses but attains the best overall performance (Acc. 20.6 in Avg@8). This suggests that verbosity alone is insufficient for physical reasoning, and that high-quality, executable code aligned with physical laws is a stronger determinant of success.

**Response Length for Trained Models** Fig. 9 shows the response length change of during RL training with different reward functions. As shown, all the lengths of three models are first decreasing, and then keeps fluctuating as the training goes on. Model initially starts at the length around $12,000$, while finally keeps relatively stable between $9,500$ and $10,000$.

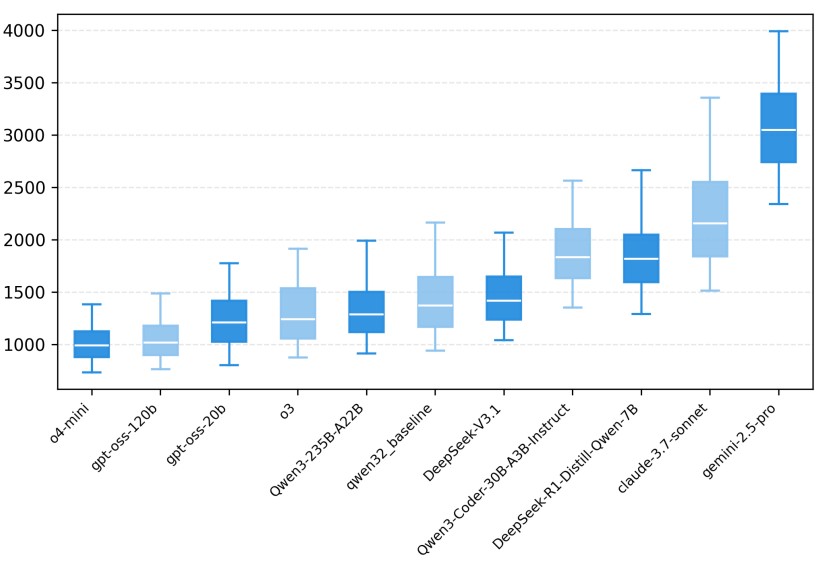

Figure 6: Distribution of response token lengths across models on the SimuPhy test set. The x-axis lists models, and the y-axis reports the average token counts of their generated answers.

# D    MORE EXAMPLES

**Mechanics**: A 4.5 kg lead block is placed on a 55-degree inclined plane inside a 2-meter-long and 1-meter-high glass chamber. The plane is made of steel with a layer of fine dust, giving a coefficient of kinetic friction of 0.25. The chamber maintains a pressure of 1 atm and a temperature of 21°C with dry air. As the block slides down, it impacts a set of small, colored plastic cones arranged at the base, knocking them over sequentially.

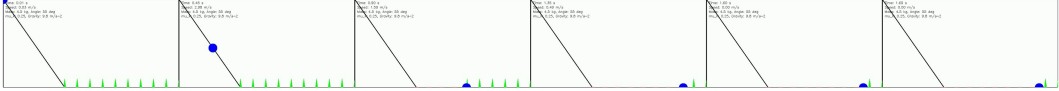

**Mechanics** : Inside a 3-meter tall glass box, a 0.15 kg steel ball is dropped from a height of 2.5 meters. The floor of the box is made of polished wood, and the ball bounces up to 1.8 meters after impact. In the box, a fan creates a gentle horizontal breeze with a constant speed of 0.5 m/s, slightly influencing the ball's rebound trajectory. This ongoing motion lasts for over 10 seconds as the ball gradually comes to a stop.

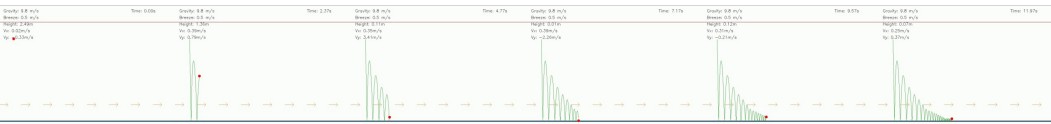

**Electromagnetism** : Inside a cylindrical glass container with a radius of 1 meter and a height of 2 meters, a solid metal ring with a radius of 0.3 meters and a mass of 2 kilograms rotates horizontally around its central axis. The inside of the container is lined with a smooth, non-conductive ceramic material. The container is filled with a uniform magnetic field of 0.05 Tesla directed vertically upwards. As the ring rotates with a constant angular velocity of 5 radians per second, it generates an electromotive force (emf) due to its motion through the magnetic field. The generated emf is measured by connecting leads to a small voltmeter attached to the ring, which is represented by a red box. The magnetic field is visualized as vertical blue lines, and the ring as a green circle.

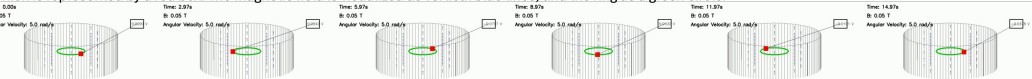

**Electromagnetism** : In a wooden box with dimensions 2 meters by 2 meters by 2 meters, a metal ring with a radius of 0.5 meters and a mass of 3 kilograms is suspended in the center by a lightweight, insulating rod. The ring rotates around a vertical axis passing through its center at an angular speed of 2 radians per second. The box contains a magnetic field of 0.08 Tesla oriented horizontally from front to back. As the ring spins, it generates an emf due to its movement through the magnetic field, which is monitored by an oscilloscope connected to the ring. The magnetic field is represented by horizontal green lines, and the ring is depicted as a red circle.

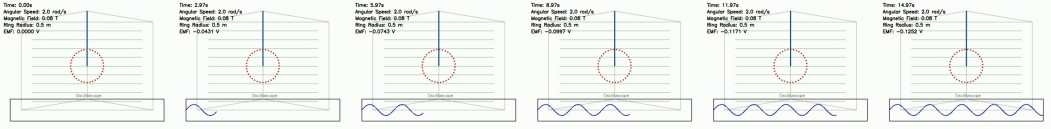

**Optics** : Within a 1-meter-long glass tube filled with glycerin (refractive index 1.47), a triangular glass prism with a refractive index of 1.52 is placed on a flat base. The tube is sealed and laid horizontally on a lab bench. A red laser beam (650 nm) passes through an entry hole at one end of the tube, hitting the prism at an angle of 60 degrees. The light disperses into its component colors, creating a rainbow pattern along the tube's bottom. The pattern remains stable and colorful for 12 seconds, allowing for a detailed observation of the dispersion effects.

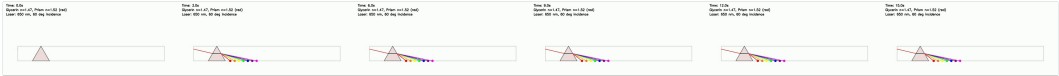

**Fluid Mechanics**: A 1 kg aluminum sphere is dropped from the top of a 2.5-meter-tall vertical glass tube. The bottom half of the tube contains a dense sugar solution, providing a visible interface and a significant increase in drag as the sphere enters. The transition from air to liquid slows the sphere noticeably, taking 16 seconds to reach the bottom while creating a vortex pattern in the sugar solution.

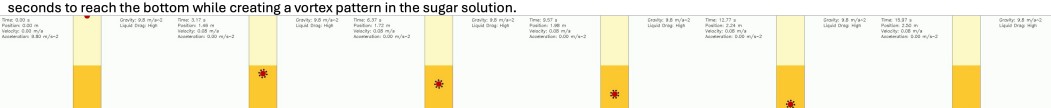

**Fluid Mechanics**: Within a large, transparent cylindrical aquarium with a diameter of 1.8 meters, a smooth plastic cylinder rotates horizontally at 22 rotations per minute. The aquarium is filled with saline water, with a density of 1.05 g/cm³. A small, floating green foam rectangle, with a mass of 0.15 kg and dimensions of 15 cm x 10 cm x 2 cm, is initially placed 80 cm from the cylinder. The rotating cylinder generates a circular current, drawing the foam inward. After 18 seconds, the foam reaches a stable path, orbiting the cylinder at a distance of 30 cm.

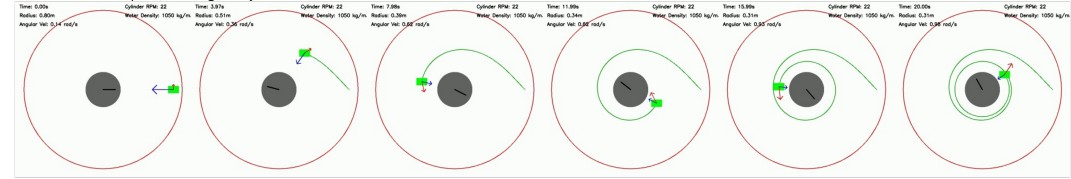

Figure 7: More representative text–to–code–to–video examples from the SimuPhy benchmark. Each video is rendered from LLM-generated code to reproduce the described dynamic physical process.

**Mechanics**: In a controlled environment, a small, yellow, 0.18 kg ball is launched horizontally from a 1.2-meter-high wooden ramp inside a steel-framed enclosure with dimensions 2m x 1m x 1.5m. The ramp's surface has a rough texture, providing friction that limits the ball's speed to 3 m/s. The air is dry, with 20% humidity. After leaving the ramp, the ball travels for 1.6 seconds before landing on a bed of rubber pellets that absorb the impact and cause the ball to bounce slightly before settling. Failure Question: The entire motion of the ball occurs inside an enclosure.

**Electromagnetism**: Inside a hemispherical dome made of clear acrylic with a diameter of 1.5 meters, a charged particle of mass 1.5 grams and charge +1.5 microcoulombs is placed at the dome's base. The dome houses a radial electric field of 600 N/C directed from the center outward. The surface inside the dome is lined with fine sandpaper to increase resistance. The humidity is maintained at 50%. When released, the particle moves along the dome's inner surface, spiraling upwards with a gradually decreasing speed due to resistance. Failure Question: The particle reaches the top of the dome after its motion.

**Optics** : A 1.5-meter-high rectangular glass tank with a refractive index of 1.5 is filled with a vegetable oil solution. A 150-gram biconvex lens with a focal length of 22 cm and a refractive index of 1.52 is submerged at the center of the tank. A yellow LED light source is positioned directly above the lens, emitting a steady beam that refracts through the lens and oil. This setup creates a luminous yellow circle that appears on the tank's bottom, maintaining its form for over 10 seconds as the light persistently interacts with the lens and oil. Failure Qustion: The yellow LED light emits a steady beam through the lens and oil.

**Fluid Mechanics**: Inside a rectangular aquarium with dimensions of 100 cm by 50 cm and filled to a height of 40 cm with a transparent gel, a small stainless steel disc with a radius of 7 cm is submerged and fixed horizontally at a depth of 20 cm. The disc is rotated at a constant speed of 22 revolutions per minute by an overhead motor. The rotation of the disc generates vortices that influence the movement of suspended fine glitter particles within the gel. Over the course of 14 seconds, the glitter particles are swept into helical paths by the vortices, gradually forming concentrated, spiral formations that trace the flow patterns within the gel, showcasing the interaction of rotational motion with a semi-solid medium. Failure Question: The glitter particles are swept into helical paths by the vortices.

**Thermodynamics**: A light beam passes through a double convex lens inside a 75 cm high, 50 cm diameter glass cylinder filled with carbon dioxide at 1.5 atm pressure. The lens, with a refractive index of 1.55 and a focal length of 20 cm, focuses the light onto a flat black metal disk, 10 cm in diameter, located 15 cm below the lens on a roughened glass base. As the light focuses, the $CO_2$ absorbs some energy, visibly forming small, localized thermal currents in the gas above the heated disk, persisting for over 12 seconds. Failure Question: A light beam passes through the double convex lens inside the cylinder.

Figure 8: Representative inconsistency cases from the VLM judgement. Each example presents a text–to–code–to–video pipeline result where the VLM flagged mismatches (highlighted in blue) between the textual description and the rendered video. These failures capture common error modes, including missing or incomplete motions, object penetration through solid boundaries, and mismatches between described and simulated dynamics.

## E    LIMITATIONS AND FUTURE WORK

SimuPhy currently relies on synthetic simulations and VLM-based evaluation, which may introduce model bias and does not yet cover the full richness of real-world physics (e.g. 3D lighting). Future work includes (1) expanding domains and realism (3D scenes, richer materials/forces, real videos with sensor data), (2) strengthening rewards via multi-VLM or human-in-the-loop adjudication, uncertainty calibration, and differentiable/analytic physics engines, and (3) releasing an open leaderboard and inference-time training baselines to foster reproducible progress. We hope SimuPhy and RLVR provide a foundation for bridging language models and simulation physical understanding.

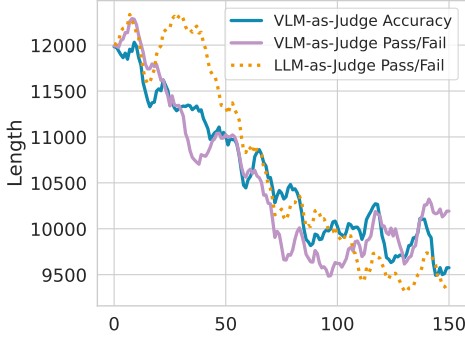

Figure 9: Model response length change during RL training with steps.

## F    USE OF LLMs IN THIS WORK

In this work, we employ large language models (LLMs) in two primary ways. First, LLMs are integrated into the data construction pipeline, enabling the automatic generation of scenarios and verification questions. Second, LLMs are used as writing assistants to help refine and polish the manuscript.

## G    PROMPT USED IN THIS PAPER

**System Prompt for Code Generation**

You are an expert computational physicist specializing in scientific visualization and simulation.

You also an excellent programmer.

Your expertise includes creating educational physics simulations that effectively communicate complex physical phenomena to diverse audiences.

**User Prompt for Code Generation**

Your task is to write a Python script that generates an educational video simulating a physical process. The video will be used in academic settings to help students better understand and visualize physics concepts. Given a textual description of a physical process:

"{content}"

Write a Python script that simulates this process and outputs a video saved as:"name.mp4", here, "name" is a user-defined parameter passed when running the script.

Requirements for the video output:

1. Use clear and distinct colors to represent different objects, trajectories, or forces.

2. Overlay a real-time timestamp that updates continuously throughout the simulation.

3. Display all relevant parameter values (e.g., gravity, speed, angle) clearly on the screen.

4. Ensure the camera view is wide enough to fully capture the entire motion, adjusting dynamically if needed. Ensure the camera view is the best view for the simulation to let the viewer see the whole process.

5. Provide smooth and continuous animation at a consistent frame rate (30 FPS).

6. Maintain a clean, uncluttered visual style with minimal distractions and a neutral background.

7. Keep the video duration between 10 and 20 seconds, slow enough to allow viewers to observe and understand the key transitions.

8. Save the output as an MP4 video in a suitable resolution (at least 360p).

9. When the process is finish, the video should finish also.

10. Use OpenCV to generate the video, and ensure the code is correct, complete, and runnable without any errors.

Focus on clarity, interpretability, and visual appeal to make the video intuitive and easy to understand for both technical and non-technical audiences.

Physics Simulation:

- Implement precise physical equations

- Use appropriate time steps for smooth motion

- Include relevant force vectors and trajectories

The final code should:

1. Initialize all necessary libraries and variables

2. Set up the video writer with specified parameters

3. Implement the physics calculations

4. Create and save the animation

5. Include error handling and resource cleanup

Ensure the code follows PEP 8 style guidelines and includes comments explaining key components. The simulation should prioritize educational value while maintaining scientific accuracy.

Instructions:

1. Output only the complete Python code. Do not include explanations or comments. The format should be like this:

```python
import cv2
import numpy as np
```

2. The code should install the dependencies in the code.

3. The code should be runnable without any errors.

4. The code should be complete and self-contained.

5. The code should be correct and accurate.

5. The code should be efficient and optimized.

6. The code should be easy to understand and modify.

---

### VLM_as_Judge Prompt

You are given a set of visual verification questions and a description of objects and motion observed in an input medium (e.g., image or video).

Your task is to evaluate each question based on whether it is correctly reflected in the visual content, considering visual cues, shape changes from viewpoint, and possible symbolic representations.

Visual Reasoning Guidelines:

1. Perspective Awareness: Objects may appear different based on viewpoint. For example:

- A cylinder may look like a circle (top view) or a rectangle/square (side view).

- A circular path may appear as a wave-like curve or straight line in 2D projection.

2. Symbolic Representations: Common simplifications may be used. You should reasonably infer their meaning:

- A series of dots or circles may represent foam markers or control points.

- A rectangle may represent a container (e.g., cylindrical viewed from the side).

- A line may represent a rubber mat or constraint boundary.

- The object and track specifics might do not match directly, if the motion can be interpreted correctly, it is still true.

- It might use color to represent different objects, such as a green line to represent the flat surface is covered with a felt-like material.

- The rotation of the object might cannot be judged from the video, but the motion can be interpreted correctly, it is still true.

3. Container Boundaries:

- If no container is drawn, you may assume the video frame itself is the container boundary.

- If a container is visible, treat it as transparent if inner content is visible.

- If the object is not visible, you should not assume it is in the container.

4. Focus on Shape & Position, not material:

- Ignore assumptions about object material, color, or texture.

- Base your decisions entirely on observable geometry (e.g., shape, layout, structure) and motion (e.g., direction, trajectory).

- Use visible movement and positioning to judge truthfulness — even if the object type is unknown.

- If the described motion is sliding down a slope, but the video shows an upward movement, the result should be "False" — regardless of material or appearance.

- Make geometric and motion-based reasoning the core of your judgment, even when objects are partially occluded.

5. Occlusion Handling:

- If an object is partially blocked, assess based on surrounding evidence whether its state or motion can still be inferred.

6. Avoid excessive uncertainty:

- If there is enough visual context and logical structure, make a confident judgment.

- Use "Not sure" only when the evidence is truly insufficient or ambiguous.

Input:

- Questions: all_questions

For each question, return:

- "index": the question index
- "question": the full question text
- "analysis": your reasoning process and visual inference
- "result": one of "True", "False", or "Not sure"
- "confidence_score": an integer from 1 (very uncertain) to 5 (very certain)

Output Format: Return a JSON list like this: [ "index": "1", "question": "The ball rolls along the circular path.", "analysis": "The object follows a closed curve consistent with a circular path from the top view.", "result": "True", "confidence_score": "5" , ... ]

