# OpenReview forum: "SimuPhy: Towards Physical Understanding, Reasoning, and Evaluation via Code Generation"
_ICLR.cc/2026/Conference — Submitted to ICLR 2026_

### Official Review · Reviewer_8Q8p · 2025-10-16

**Soundness:** 3
**Presentation:** 3
**Contribution:** 3
**Rating:** 4
**Confidence:** 3

**Summary:**

This paper introduces SimuPhy, a novel benchmark and dataset designed to evaluate and improve the ability of LLMs to understand, reason about, and simulate physical laws through text-to-code generation. The authors construct a dataset of 7,625 dynamic scenarios across five physics domains using an automated pipeline. Each scenario is paired with visual verification questions, and the generated code is executed to produce videos, which are then evaluated by a VLM for physical consistency. The authors also propose a reinforcement learning approach with verifiable rewards (RLVR) that leverages VLM-based judgments to align model outputs with physical correctness. Experiments on 10 state-of-the-art LLMs reveal significant limitations in physical reasoning, with the best model achieving only 20.6% accuracy. Through fine-tuning and RL, the authors demonstrate that a 7B model can achieve performance competitive with much larger models.

**Strengths:**

1. Combine text-to-code generation with visual verification for physical reasoning. The RLVR framework is innovative and leverages VLM judgments without human labels.

2. The paper is well-organized, with clear explanations of methodology, metrics, and results.

3. Provides a new benchmark for physical reasoning and a training methodology that improves model alignment with physical laws.

**Weaknesses:**

1. The synthetic dataset may not fully capture the complexity of real-world physics, such as 3D interactions and dynamic lighting effects. Furthermore, it primarily consists of one-dimensional cases, which limits its scope.

2. While the VLM judge is effective, its judgments may inherit biases from its base model. A more fundamental concern is the potential circularity of using a VLM's own physical reasoning capability as a reward signal to further improve itself via reinforcement learning.

3. The dataset underrepresents certain physical concepts, such as optics, which could hinder the model's ability to generalize to these domains.

4. The RL training process is computationally intensive, requiring 576 GPU-hours, which may limit the accessibility and reproducibility of this approach.

**Questions:**

See weaknesses.

---

> ### Author Response · Authors · 2025-12-03
>
> **Limited Physical Realism in Synthetic Dataset:** We thank the reviewer for this constructive comment. We agree that SimuPhy currently focuses on simplified physical settings—such as rolling, bouncing, circular motion, and basic refraction/reflection—yet even these cases remain highly challenging for frontier LLMs. As shown in Fig. 8, models frequently fail to reproduce physically consistent trajectories in these “1D–2D” scenarios. While our rendered videos depict a single view, the VLM-based verification explicitly considers 3D consistency (e.g., different viewing angles where circular motion may appear elliptical). We fully acknowledge that richer 3D environments, lighting effects, and physics engines would further improve realism, and we plan to incorporate 3D physics engines and multi-view rendering in future extensions to enhance physical and visual fidelity.
>
>
> **Biased judgment:** Thank you for raising this point. We agree that VLM-based judges may inherit biases and that using a model’s own physical reasoning as a reward signal can introduce circularity. In our setup, the reward comes from an independently trained VLM rather than the policy model itself, which reduces this issue. We will further mitigate potential circularity by incorporating human evaluation and multi-VLM voting in future work.
>
>
> **Underrepresented concepts:** We thank the reviewer for the observation. As shown in Table 4, SimuPhy includes 836 optics-related scenarios (≈11% of the dataset), covering six distinct optics concepts. Although this is fewer than the 24 mechanics concepts, the relative proportion of optics data is not small. The difference reflects the natural variability in visualizability across physical domains—mechanics and electromagnetism contain more phenomena that can be readily expressed through observable motion or field interactions, whereas areas like quantum mechanics or spacetime curvature are inherently difficult to represent via motion trajectories. Nonetheless, we agree that expanding underrepresented areas such as optics is valuable and plan to include more complex optical and wave-based scenarios in future releases.
>
>
> **GPU hours:**  We thank the reviewer for this valid concern. The reported 576 GPU-hours corresponds to training a 7B-parameter model with our RL setup, which is comparable to standard practice for reinforcement learning on models of this scale. For reference, recent 7B RL-based reasoning models such as DeepSeek-R1-Distill-Qwen-7B and Open-R1 7B report similar or higher computational costs (typically 400–800 GPU-hours) under comparable batch sizes and training steps. Our training setup is therefore within the normal range for large-model RL research. Moreover, all code, prompts, and configuration details are provided to ensure reproducibility, and lighter versions (e.g., smaller models or reduced-step RL) can be used for accessible replication without altering the overall methodology.

---

### Official Review · Reviewer_ATNw · 2025-10-29

**Soundness:** 3
**Presentation:** 3
**Contribution:** 2
**Rating:** 4
**Confidence:** 4

**Summary:**

This paper introduces SimuPhy, a novel benchmark and training methodology designed to address the significant limitations of Large Language Models (LLMs) in reasoning about dynamic physical processes. The authors argue that existing models fail to generate code that can accurately and consistently simulate physical scenarios described.

**Strengths:**

- A Novel Evaluation Pipeline: The core of their approach is a "text-to-code-to-video-to-VLM adjudication" loop. An LLM's generated Python simulation code is executed, rendered into a video, and then evaluated by a Vision-Language Model (VLM) judge (Qwen-2.5-72B-VL) against a set of verification questions.
- Comprehensive Experiments: Authors revaluated 10 advanced LLMs and provided detailed experiments

**Weaknesses:**

- Circular Logic in Benchmark Design: The paper's most fundamental flaw is the "closed-loop" nature of its benchmark. The data is AI-generated (by GPT-4o), AI-validated (by DeepSeek-R1), and AI-judged (by Qwen-VL). This "AI-generation, AI-verification, AI-adjudication" system raises immediate concerns about circularity. It is unclear whether the benchmark measures an LLM's "genuine physical understanding".
- Misleading and Unfair Comparisons: A key conclusion—that the authors' trained 7B model (45.0% Pass@8, Table 3) surpasses powerful models like Gemini-2.5-pro (40.0% Pass@8, Table 1)—is methodologically unfair and misleading. The authors' 7B model was specifically trained (both SFT and RLVR) on the SimuPhy dataset itself. The frontier models (Gemini, Qwen, etc.) were evaluated in a zero-shot setting without any specific training on the SimuPhy domain or reward signal. This is a clear case of "training on the test set's domain." This "victory" does not prove the 7B model has superior, generalizable physical reasoning. It only proves that it can be effectively optimized to "take the test" and "please the referee" (the VLM) for this specific benchmark.

**Questions:**

- Have you tested the 7B model (trained with SFT + RLVR) on other physics benchmarks (e.g., text-based QA like PhysBench or other simulation tasks)? This would be essential to determine if the "understanding" gained by optimizing for the SimuPhy VLM judge is generalizable or if it is an overfitted, task-specific skill.

---

> ### Author Response · Authors · 2025-12-03
>
> **Circular Logic in Benchmark Design:** Thank you for highlighting this concern. We agree that using AI-generated data, AI-based verification, and AI judges can raise questions about circularity. Our goal, however, is to build a scalable framework for evaluating code-based motion reconstruction. The data generation, verification, and judgment come from different models trained on different sources, reducing direct circular reinforcement. Moreover, our human evaluation shows a 93.2% agreement with the VLM judge, supporting the reliability of the setup. Still, we acknowledge that AI-only pipelines may introduce shared biases. In future work, we will incorporate more human evaluation and multiple independent VLM judges to further mitigate circularity.
>
>
> **Misleading and Unfair Comparisons:** Thank you for pointing this out. We agree that comparing our 7B model—trained on SimuPhy through both SFT and RLVR—with frontier models evaluated in a purely zero-shot setting does not demonstrate superior generalizable physical reasoning. Our intention is different: the experiments aim to show that a smaller model, once trained on our dataset, can reach or exceed the zero-shot performance of much larger models on this task. This indicates that current frontier models are not yet optimized for code-based motion reconstruction and that their performance can be further improved through targeted training—a trend also observed in related work such as AgentTuning [1]. We will clarify this intent and avoid framing the result as evidence of inherent superiority of the 7B model.
>
> [1] Zeng, Aohan, et al. "Agenttuning: Enabling generalized agent abilities for llms." Findings of the Association for Computational Linguistics: ACL 2024. 2024.
>
>
> **other physics benchmarks:** Thank you for the question. Assessing cross-benchmark generalization is indeed important, and we plan to explore evaluations on broader physics reasoning and simulation datasets in future work.

---

### Official Review · Reviewer_r5GR · 2025-10-29

**Soundness:** 2
**Presentation:** 2
**Contribution:** 1
**Rating:** 2
**Confidence:** 4

**Summary:**

The paper introduces SimuPhy, a text-to-code-to-video benchmark testing LLMs’ physical reasoning. Models generate Python code from motion descriptions to render videos verified by a VLM. The dataset spans 7,625 physics scenarios. Top models reach 54.7% Pass@8. Using reinforcement learning with vision-verifiable rewards (RLVR), a 7B model achieves 45.0% Pass@8, showing progress toward physical understanding.

**Strengths:**

- Substantial dataset spanning five classical physics domains, plus a curated, human-checked test split.
- Practical evaluation pipeline with multiple operational metrics (Executable, Render, Playable, Accuracy), revealing distinct failure modes beyond simple correctness.
- Broad baseline sweep over 10 strong LLMs, the low accuracies highlight a real gap and the benchmark’s difficulty.
- RL with verifiable vision reward is a sensible training strategy that improves scenario-level Pass@8 notably for a small model, reward ablations (VLM-acc, VLM-binary, LLM-as-judge) are informative.
- Initial judge-validation against human labels provides some evidence that the VLM judge is usable.

**Weaknesses:**

- Physical grounding is weak: the task permits hand-crafted 2D OpenCV animations without any requirement to use a physics engine or enforce conservation/constraints, so “simulation of physical laws” can degenerate into scripted drawing that merely looks plausible. This risks measuring graphics scripting rather than physical reasoning. Without constraining the toolset to bona fide physics engines (e.g., PyBullet, Box2D, MuJoCo, differentiable optics/EM solvers) or quantitatively checking laws, the task primarily assesses the model’s ability to draw plausible trajectories. Qualitative VLM questions cannot distinguish a kinematically scripted spiral from one generated by correct force integration. This undermines the stated goal of measuring "physical-law reasoning."

- Single-judge dependence and potential reward hacking: all core results depend on one VLM (Qwen-2.5-72B-VL) and a pass rule that treats low-confidence “Not sure” as correct (with up to two allowed). RL directly optimizes this judge, making overfitting to its idiosyncrasies or ambiguity-seeking behavior likely. No cross-judge or multi-judge aggregation is reported for the final scores.

-Limited human evaluation where it matters most: the 93.2% agreement is reported on pre-RL data. There is no targeted human audit of the RLVR-trained model’s outputs to check for reward hacking or degradation in true physical fidelity.

- Evaluation rule design is permissive and may inflate scores: counting low-confidence “Not sure” as correct and allowing up to two such answers to pass can be exploited, there is no sensitivity study showing that results are stable when tightening this criterion.

- Small test set and class imbalance: only 300 test items across 52 concepts and 5 domains, with long-tailed coverage.

- Even by the paper’s own numbers, consistent single-sample performance remains weak (Avg@8), again suggesting that improvements are largely in best-of-N sampling rather than genuine robustness.

- Novelty of “verifiable rewards” is incremental: RL from automated judges (including VLMs) is rapidly becoming standard.

My recommendation: Reject. Key reasons: (1) The benchmark’s core task allows non-physical, scripted animations to pass, so the central claim of advancing “physical law understanding” is not convincingly supported (2) Heavy reliance on a single VLM judge and permissive pass criteria invites reward hacking and overfitting, with no cross-judge/human verification of the RL improvements.

**Questions:**

- What prevents a model from simply scripting 2D animations that look like the described motion without integrating the relevant equations of motion or laws (e.g., Snell’s law, Lorentz force)? Can you enforce the use of physics engines or provide automated checks for conserved quantities where applicable? Add constrained baselines that must use a designated physics engine per domain, plus a “pure animation” baseline. This would directly test whether the benchmark discriminates "physics-based" simulation from scripted graphics.

- Do you have any quantitative verifications (e.g., energy/momentum preservation in elastic collisions, angle ratios for refraction) on a subset of scenarios?

---

> ### Author Response · Authors · 2025-12-03
>
> **Physical grounding:** Thank you for the comment. We intentionally focus on qualitative motion because current LLMs fail basic kinematic consistency and often generate implausible behaviors despite high render rates, such as penetrating solids and inverse motion. We agree that physics-engine baselines and quantitative checks would strengthen grounding and plan to explore them.
>
>
> **Reward hacking:** Thank you for raising this concern. We acknowledge the risk and reward hacking in a single VLM judge. Although we could not include human evaluation of the RL-trained model now, we plan to add human assessments and adopt a multi-VLM voting scheme in future work to mitigate these issues.
>
>
> **Human evaluation:** Thank you for your suggestion. We agree and will incorporate more in future.
>
>
> **Evaluation rule design is permissive and may inflate scores:** Thank you for the feedback. We allow up to two low-confidence “Not sure” answers because this yields the highest alignment with human evaluation (93.2%). Even treating all “Not sure” cases as false still preserves strong agreement (90.7%). We will explore stricter and more robust handling of “Not sure” predictions in future work.
>
>
> **Small test set and class imbalance:** We appreciate the reviewer’s concern. Our 300-example test set is indeed modest, but it is fully human-verified (87% inter-annotator agreement) and represents a practical balance between diversity, verification cost, and reliability. For comparison, several established benchmarks—such as AIME 2025 (15 test items) and GPQA (448 examples)—are of similar scale.
>
>
> **Single-sample performance remains weak:** Thank you for the comment. The weaker single-sample performance reflects the difficulty of our task: models must generate executable Python code to reconstruct a physical scene, rather than provide a direct answer. This code-based visual simulation setting is largely unexplored and not what existing models are optimized for, leading to low one-shot accuracy. The improvements from best-of-N sampling show that models have the required reasoning ability but often need multiple attempts to satisfy all motion and physical constraints.
>
>
> **Novelty:** Thank you for pointing out these related works. VLM-based reward generation is rapidly evolving, and we group prior approaches as follows:
>
> *VLM Preferences & Zero-Shot Rewards:* Methods like RL-VLM-F [1] and others [2] demonstrate that CLIP-style models can serve as zero-shot reward models by computing similarity or preferences over observation pairs conditioned on language goals.
> Video-Level Alignment: RLAIF-V [3] and Video-RLAIF [5] extend this to video generation, using VLM feedback for preference learning and alignment at the video level.
>
> *Enhanced Feedback Mechanisms:* Recent methods such as VARP [4] and Code-as-Reward [6] improve VLM-based feedback, but our method differs fundamentally in action space, verification mechanism, and objective. Code → Video → VLM Loop: Unlike prior works where the VLM judges external state observations or pixel outputs, our policy generates executable code. The "judge" does not merely score text or static similarity; it scores the rendered dynamics resulting from that code. Execution-Coupled Reward: Our reward depends on both the VLM output and the executability of the generated code. Code that fails to compile or render receives zero reward, forcing the policy to learn semantic and syntactic consistency simultaneously—unlike [6], where the VLM only specifies the reward. Scenario-Specific Verification: Instead of generic preference scores, our VLM judge answers decomposed, scenario-specific verification questions with uncertainty handling. To our knowledge, we are the first to train a language model for physical reasoning via this simulation-in-the-loop verification.
>
> *References*
>
> *[1] Rocamonde, J., et al. "Vision-Language Models are Zero-Shot Reward Models for Reinforcement Learning." ICLR, 2024.*
>
> *[2] Baumli, K., et al. "Vision-Language Models as a Source of Rewards." arXiv:2312.09187, 2024.*
>
> *[3] Yu, X., et al. "RLAIF-V: Open-Source AI Feedback Leads to Super GPT-4V Trustworthiness." CVPR, 2025.*
>
> *[4] Singh, A., et al. "VARP: Reinforcement Learning from Vision-Language Model Feedback..." arXiv:2503.13817, 2025.*
>
> *[5] Ahn, D., et al. "Tuning Large Multimodal Models for Videos using Reinforcement Learning from AI Feedback." ACL, 2024.*
>
> *[6] Wang, K., et al. "Guiding reinforcement learning with shaping rewards provided by the vision-language model." Engineering Applications of Artificial Intelligence, 2025.*
>
>
> **Physics-Based Simulation:** We thank the reviewer for the suggestion. SimuPhy focuses on qualitative motion trends because LLMs fail on simple issues like boundary penetration or reversed motion. We agree that adding physics-engine and animation baselines, along with automated quantitative evaluations, would strengthen grounding, and we plan to explore these in future work.

---

### Official Review · Reviewer_9j6y · 2025-10-31

**Soundness:** 2
**Presentation:** 3
**Contribution:** 2
**Rating:** 4
**Confidence:** 4

**Summary:**

The paper introduces SimuPhy, a large-scale benchmark and dataset designed to evaluate large language models’ (LLMs) ability to understand and simulate physical processes through code generation. In SimuPhy, an LLM receives a natural-language description of a motion scenario (e.g., “a spinning disk on a frictionless table”) and must generate Python code that produces a corresponding simulation video. The resulting video is then automatically evaluated using a vision–language model (VLM), which answers predefined verification questions to determine physical consistency between the textual description and the generated simulation. The authors evaluate 10 state-of-the-art LLMs, finding that even the strongest model (DeepSeek-R1-0528, 671B) achieves only 20.6% accuracy. The paper also introduces the RLVR scheme, which uses visual verification signals from a VLM as training feedback. Combined with supervised fine-tuning, RLVR substantially improves model performance: a 7B model trained with this approach reaches 45% pass rate.

**Strengths:**

- The benchmark has a large scope, covering a wide range of physics domains and dynamic scenarios.
 - The study includes comprehensive experiments with multiple leading LLMs, highlighting consistent performance patterns across models.
 - The availability of the dataset and the RLVR baseline supports reproducibility and provides a foundation for future research.

**Weaknesses:**

-  It is unclear whether the benchmark truly measures physical understanding or rather the ability to produce correct simulation code. Even a model (or human) with solid conceptual understanding could fail due to coding mistakes such as axis misalignment, suggesting the latter may dominate. The actual human score would reveal that.
 - The data generation pipeline heavily depends on multiple LLMs for scenario creation and filtering, yet the paper’s own results indicate that current LLMs struggle with physical reasoning.
 - The VLM-as-a-judge component is vulnerable to reward hacking and may produce inconsistent or biased judgments; prior works have shown that visual verification models can be exploited or yield false positives.
 - Since Avg@8 effectively corresponds to pass@1 with eight samples used for estimation, adopting standard pass@k notation would improve clarity and comparability.

**Questions:**

- Could you clarify which parts of the data pipeline underwent human evaluation? Was it applied only to the verification questions?
 - Have you considered conducting a human evaluation of the RLVR-trained model outputs to verify that improvements reflect genuine physical correctness rather than reward hacking or overfitting to the VLM judge?
 - If it is possible, please provide the human score for comparison.

---

> ### Author Response · Authors · 2025-12-03
>
> **SimuPhy might measure coding accuracy rather than physical understanding:** We thank the reviewer for this insightful comment. As noted in lines 50–51, SimuPhy is designed to evaluate both aspects. Table 1 shows models achieving high render rates (DeepSeek-R1 96.8%, Qwen3-235B 95.1%) but markedly different accuracies (20.6% vs. 12.3%), indicating that physical understanding—not coding errors—is the main bottleneck. Moreover, human validation yields 87% inter-annotator agreement and 93.2% consistency with the VLM-based evaluator, confirming that our automatic evaluation reliably reflects human judgments of physical plausibility. Together, these results demonstrate that SimuPhy primarily measures physical understanding rather than coding precision.
>
>
> **Overreliance on LLMs with Limited Physical Reasoning Capability:** We appreciate the reviewer’s thoughtful comment. While our pipeline uses LLMs for scenario creation and our results show LLMs struggle with physical reasoning, this reflects a crucial distinction: step-level generation vs. end-to-end reasoning. Each stage of our pipeline—domain selection, scenario and verification question generation, quality improvement, and training data generation—relies on standard LLM capabilities such as text synthesis, consistency checking, and plausibility filtering, which current models perform reliably. The real challenge lies in the end-to-end task of translating a physical description into executable code that produces a physically consistent simulation. For example, even for a simple “ball rolling down a slope,” many frontier models still generate code where the ball moves against gravity.  This gap is precisely why SimuPhy is valuable: while LLMs can assist dataset construction through modular, verifiable steps with human oversight (87% inter-annotator agreement, Sec.3.2), the benchmark itself demands fundamentally deeper physical understanding that current models lack.
>
>
> **Reward hacking:** Thank you for highlighting the potential vulnerability of the VLM-as-a-judge component to reward hacking and the risk of inconsistent or biased judgments. We acknowledge that prior work has shown that visual verification models can be exploited or may produce false positives. Due to time constraints, we were not able to include human evaluation of the RL-trained model in the current submission. As part of future work, we will conduct human assessments to verify whether reward hacking occurs in practice. We also plan to adopt a multi-VLM voting scheme to reduce the likelihood of reward hacking further and mitigate biases from any single model.
>
>
> **Standard pass@k Notation for Clarity and Comparability:** We thank the reviewer for the helpful suggestion. In our paper, Avg@8 and Pass@8 are defined differently to capture complementary aspects of model performance. Avg@8 represents the average accuracy across eight independent generations for each sample, reflecting the model’s overall reliability. Pass@8, on the other hand, follows the standard pass@k definition—counting a case as correct if at least one of the eight generations is valid—measuring best-case performance under multiple attempts. We agree that clarifying this distinction would improve clarity and comparability, and we will revise the text to explicitly explain these definitions and their relation to conventional pass@k notation.
>
>
> **Human evaluation:** We appreciate the reviewer’s question. The human evaluation in SimuPhy was primarily conducted on the alignment between the input scenario description and the LLM-generated video simulation, rather than on the verification questions alone. Specifically, human annotators watched the rendered videos and judged whether the simulated motion correctly reflected the described physical scenario (e.g., whether a ball “rolling down a slope” in text indeed moved downhill in the video). This evaluation directly measures the scene-level consistency between textual descriptions and generated dynamics, serving as a ground-truth check for the VLM-based judgments. As reported in Section 3.2, two annotators achieved 87% inter-annotator agreement, and the final human consensus achieved 93.2% consistency with the VLM-as-judge results. This confirms that our human evaluation focused on verifying physical alignment rather than solely checking the verification questions.
>
>
> **RLVR outputs human evaluation:** Thank you for pointing out. We will report this in future work.
> Human score: Reported in Section 3.2 with 87% inter-annotator agreement and 93.2% consistency with the VLM-based evaluator; the human score for RLVR will be reported in future work.

---

### Author Response · Authors · 2025-12-03

We thank the reviewers for their insightful and constructive comments on our work. We address each question and concern point by point below.

---

### Meta-Review · Area_Chair_HEEV · 2026-01-08

**Summary:**

The paper received mixed but overall negative evaluations, with scores ranging from 2 to 4, and one reviewer explicitly recommending rejection. While reviewers acknowledged the ambition and scale of SimuPhy, as well as the clarity of the text-to-code-to-video evaluation pipeline, they consistently raised fundamental concerns about the validity of the benchmark and the strength of the claims. The most significant issues include weak physical grounding—since the task allows scripted, visually plausible animations without enforcing physical laws or the use of physics engines—heavy reliance on a single VLM judge with permissive evaluation rules, and the resulting risk of reward hacking or overfitting. As a result, the central claim that SimuPhy meaningfully measures and advances physical-law understanding in LLMs is not convincingly supported in its current form. Given that these weaknesses outweigh the contributions at this stage, I recommend Reject.

**Reviewer Concerns:**

Several reviewers also pointed out circularity in the benchmark design (AI-generated data, AI-based verification, AI-based adjudication), limited human evaluation where it matters most (especially for RL-trained outputs), small and imbalanced test sets, and potentially misleading comparisons between a SimuPhy-trained 7B model and large frontier models evaluated only zero-shot.

**Reviewer Scores:**

The paper received overall negative evaluations, with scores ranging from 2 to 4: 2,4,4,4.

---

### Decision · Program_Chairs · 2026-01-26

Reject